

# Microbial growth in biobeds for treatment of residual pesticide in banana plantations

Verónica I. Domínguez-Rodríguez[1], Eduardo Baltierra-Trejo[1,2], Rodolfo Gómez-Cruz[1] and Randy H. Adams[1]

[1] División Académica de Ciencias Biológicas, Universidad Juárez Autónoma de Tabasco, Villahermosa, Tabasco, Mexico

[2] Catédras CONACyT, Consejo Nacional de Ciencia y Tecnología, Mexico City, Mexico

## ABSTRACT

**Background.** High doses of ethylenebisdithiocarbamate (EBDC) are used in banana production, and unused pesticide mixture (solution) is often disposed of improperly. This can result in soil and water contamination and present an undue risk to rural communities and the environment. An alternative to reduce the environmental impacts caused by pesticide residues is the biobeds treatment. It is necessary to establish if the composition of the proposed biomixtures supports microbial activity to degrade pesticides in biobeds. This research aimed to evaluate the EBDC effect on the distribution and abundance of microbial populations in polluted biomixtures .

**Methods.** For this purpose, a biomixture based on banana stem, mulch, and Fluvisol soil (50:25:25% v/v) was prepared and polluted with 1,000 mg L$^{-1}$ EBDC. The response variables kinetics were determined every 14 days for three months, such as pH, organic matter, moisture, cation exchange capacity, microbial colonies, and cell counts at three depths within the experimental units.

**Results.** EBDC reduced the number of microbial colonies by 72%. Bacterial cells rapidly decreased by 69% and fungi 89% on the surface, while the decrease was gradual and steady at the middle and bottom of the biobed. The microbial populations stabilized at day 42, and the bacteria showed a total recovery on day 84, but the fungi slightly less. At the end of the experiment, the concentration of EBDC in the biomixture was 1.3–4.1 mg L$^{-1}$. A correlation was found between fungal count (colonies and cells) with EBDC concentration. A replacement of the biomixture is suggested if the bacterial population becomes less than $40 \times 10^6$ CFU mL$^{-1}$ and the fungal population less than $8 \times 10^4$ CFU mL$^{-1}$ or if the direct cell count becomes lower than $50 \times 10^4$ cells mL$^{-1}$ in bacteria and $8 \times 10^2$ cells mL$^{-1}$ in fungi.

**Conclusion.** The biomixture based on banana stem supports the microbial activity necessary for the degradation of the EBDC pesticide. It was found that fungi could be used as indicators of the pollutant degradation process in the biomixtures. Microbial counts were useful to establish the mobility and degradation time of the pesticide and the effectiveness of the biomixture. Based on the results, it is appropriate to include the quantification of microbial populations to assess the effectiveness of pesticide degradation and the maturity level of the biomixture.

Corresponding author
Eduardo Baltierra-Trejo,
eduardo.baltierra@conacyt.mx

## INTRODUCTION

It is estimated that 155 million tons of bananas are produced annually in tropical regions of the world (*FAO, 2019*). Humid conditions and high temperatures favor the appearance of pests. The most frequent and damaging is the black Sigatoka fungus. (*Mycosphaerella fijiensis* Morelet) (*Drenth & Guest, 2016*). The fungicide ethylenebisdithiocarbamate (EBDC), commercially known as Mancozeb, is applied at weekly doses of 2 kg ha$^{-1}$ throughout the year to maintain intensive production. EBDC has a short half-life in the environment, but it degrades by photooxidation into ethylenethiourea (ETU), a recalcitrant compound with mutagenic and carcinogenic potential (*Gupta, 2018*).

ETU is mobilized in the environment due to spills and inappropriate practices during the filling application equipment, thus contaminating soil and water (*Morillo & Villaverde, 2017*). It has been found that the concentration of ETU in wastewater generated in banana plantations is as high as 800 mg L$^{-1}$ (*Domínguez et al., 2015*; *Geissen et al., 2010*). As a result, workers and inhabitants in banana plantations may be exposed to acute poisoning and chronic degenerative diseases (*Rea & Patel, 2017*). It is estimated that three million acute pesticide poisoning cases occur each year worldwide in agricultural areas, of which 10% are fatal (*Mew et al., 2017*). Therefore, it is necessary to develop strategies to mitigate the impact of pesticides.

An alternative for treating pesticide residues is to adsorb and degrade them in a construction known as biobed. According to the original model proposed in Sweden, the biobed is filled with organic substrates or biomixture, composed of soil, peat from swamps, and wheat straw (*Torstensson, 2000*). In the Swedish biomixture, peat from swamps is the primary source of microorganisms, while wheat straw is a source of carbon and lignin, stimulating fungi enzymatic activity (*Castillo & Torstensson, 2007*). The useful life of the Swedish biomixture has been estimated at 6–8 years (*Torstensson, 2000*).

Peat and wheat straw are difficult to obtain near banana-producing areas, but locally available materials could be used in biomixtures production (*Vischetti et al., 2007*). In this respect, alternatives have been investigated to replace peat with another material with pollutant sorption capacity. For example, *Vischetti et al. (2007)* evaluated the use of biomixtures prepared with composts, *Gao et al. (2015)* spent mushroom substrate, and *Mukherjee et al. (2016)* with biochar. Other researches have evaluated the use of alternative sources of lignin to replace straw. For example, *Karanasios et al. (2010)* found promising results in corn cobs, sunflower residues, grape stalks, orange peels, olive tree pruning, and citrus peel; while *Domínguez et al. (2021)* used sugarcane tip, eucalyptus chip, and banana stem.

Therefore, it is necessary to evaluate the degradability and useful life of the proposed biomixtures *Dzionek, Wojcieszyńska & Guzik (2016)*. Among the parameters proposed for monitoring the biomixtures are the moisture, pH, cation exchange capacity, organic matter, carbon, and nitrogen content (*Delgado et al., 2019*; *Karanasios et al., 2010*); and biological variables such as enzyme activity, respiration, and microbial biomass (*Adak et al., 2020*; *Vischetti et al., 2007*). In this regard, the biological factors involved have been less addressed.

For example, *Vischetti et al. (2007)* compared the degradation of the chlorpyrifos pesticide at a concentration of 50 mg kg$^{-1}$ in a biobed with two different biomixtures, one with peat and the other with compost mixed with vine pruning and soil. The biomixtures with peat showed a higher pesticide degradation than the compost. Furthermore, in the biomixtures, the pesticide inhibited the respiratory activity by 50% and the microbial biomass by 60%, while with peat, there was no inhibition. The biomixture with peat had a higher abundance of microbial species, mainly fungi. They concluded that fungal diversity was related to the pH and higher carbon content.

Subsequent studies have considered the influence of environmental factors on microbial activity during the degradation of pesticides in the biomixtures. For example, *Castro et al. (2017)* evaluated the effect of carbofuran (20 mg kg$^{-1}$) on microbial species diversity in a biomixture at 25 °C. It was determined that the biomixture gradually lost its effectiveness, and only 88% of the pesticide was degraded in 180 days. It was concluded that species diversity varied mainly due to the biomixture aging and secondarily due to the pesticide. The useful life of the biomixture at 25 °C was considered to be one year.

On the other hand, *Adak et al. (2020)* evaluated the effect on microorganisms of imidacloprid (178 mg kg$^{-1}$) in a biomixture prepared with straw, manure, and soil (2:1:1 v) in a tropical climate. After 90 days, 95% of degradation was achieved. The pesticide degradation was related to fluorescein diacetate hydrolase and dehydrogenase enzyme activities, but not to β-glucosidase. It was concluded that fungi were less affected than bacteria by the pesticide.

Therefore, research evaluating the relationship of physicochemical and biological parameters in different biomixtures, and climates is of interest. The microbiological studies would identify the effectiveness or depletion of the biomixture, the accumulation or mineralization of toxic compounds, and the mobility of pesticides in the biobed (*Vareli et al., 2018*). In this sense, previous research has not considered microbial colony and cell counts. The research aim of the present study was to evaluate the effect of EBDC on the distribution and growth of fungi and bacteria in a biomixture prepared with materials available in a banana plantation under warm-humid conditions. Therefore, a biomixture based on banana stem, soil, and mulch was prepared and polluted with 1,000 mg L$^{-1}$ EBDC. Subsequently, the degradation into ETU was evaluated as well as the effect on microbial colonies and cells at three depths within the experimental units.

## MATERIALS & METHODS

The toxic effect of the EBDC on microbial distribution and abundance in simulated biobeds was evaluated in a tropical-humid environment. The biomixture used was based on banana stem, mulch, and soil. The degradation kinetics of the pesticide was performed for three months, as described below:

*Preparation of the biomixture and experimental units.* The materials required for biomixture, such as banana stem, mulch (top "O" layer of soil formed mainly by decomposing leaf litter in the banana plantation), and soil (Fluvisol) from a depth of 25 to 50 cm, were collected at Ranchería Miahuatlán, Tabasco (longitude: 18.020500,

**Table 1   Initial characterization of the materials used in the preparation of the biomixture.**

|  | Soil | Banana stem | Mulch | Biomixture |
|---|---|---|---|---|
| Texture (%) | 54 clay<br>31 silt<br>15 sand | – | – | – |
| Field capacity (%) | 35.0 | – | – | 41.2 |
| Moisture (%) | 45.6 | 65.2 | 12.5 | 55.3 |
| pH | 5.5 | 10.1 | 5.7 | 6.7 |
| Electrical conductivity (dS m$^{-1}$) | 0.04 | 0.06 | 0.1 | 0.08 |
| Cation exchange capacity (cmol(+) kg$^{-1}$) | 58.2 | 28.2 | 12.3 | 86.2 |
| Organic matter content (%) | 6.5 | 33.2 | 17.2 | 24.4 |
| Carbon (%) | 24 | 74 | 51 | 48 |
| Nitrogen (%) | 0.25 | 0.87 | 0.98 | 1.85 |
| Ratio Carbon-nitrogen (C/N) | 96 | 85 | 1 | 25 |

latitude: −93.297000). The banana stem was chopped into fragments of approximately 3 × 1 cm. Banana stem, mulch, and soil were mixed in the ratio of 50:25:25% (v/v). The biomixture was composted for 50 days before the pollution. The physicochemical characterization of the soil and the materials used in the preparation of the biomixture was carried out. The results are shown in Table 1.

Biobeds were simulated in laboratory-scale experimental units built with polyethylene cylinders (length 50 cm, diameter 9.5 cm). The experimental units were buried at ground level to simulate field conditions and kept in an outdoor patio adjacent to the laboratory. During the study period, the ambient temperature was approximately 20–35 °C. The arrangement of the units was randomized. Manual irrigation was performed twice a week, with 300 mL of water per unit, evenly spreading the liquid on the surface of the biobeds.

Four treatments were evaluated: unpolluted and polluted biomixtures, unpolluted and polluted soils. Polluted treatments were irrigated at the beginning of the experiment with 1,000 mg L$^{-1}$ EBDC (Mancozeb®).

Openings of five cm in diameter were made to sample the experimental units at three depth levels: surface (5–10 cm), middle (25–30 cm), and bottom (45–50 cm). The samples were taken from the units with a spoon twice a week for 84 days to evaluate physicochemical parameters: field capacity, pH, humidity, cation exchange capacity, organic matter content, and pesticide concentration, as well as biological parameters: number of microbial colonies and cell count in suspension. The samples of each variable were compared at three levels of depth throughout the experimental unit.

*Soil texture.* The hydrometer or Bouyoucos method was used to determine the soil texture. 30% $H_2O_2$ was added to 60 g of a soil sample to oxidize the organic matter. Water and 10 mL of sodium hexametaphosphate $(NaPO_3)_6$ were added to 50 g of disaggregated sample. The components were mixed for 5 min. The mixture was topped-off at 1,000 mL with distilled water, stirred for 1 min, and analyzed at 40 s and 2 h with the hydrometer and thermometer (*Pansu & Gautheyrou, 2007*).

*Field capacity (FC).* 100 g of dry sample was moistened with water. Afterwards, the wet sample was drained for 24 h and weighed. It was then dried at 60 °C for 24 h (Kirkham, 2014; *Pansu & Gautheyrou, 2007*). The field capacity was calculated with Eq.(1):

$$\% \text{ FC} = \frac{\text{wet weight (g)} - \text{dry weight (g)}}{\text{wet weight (g)}} \times 100 \tag{1}$$

*Total organic carbon.* This was determined in the materials used to prepare the biomixtures, using the Walkley-Black method (*De Vos et al., 2007*). To 0.5 g of soil was added five mL of $K_2Cr_2O_7$ and 10 mL of $H_2SO_4$. The mixture was let digests for 30 min, and five mL of $H_3PO_4$ and 100 mL of water were added. Finally, it was titrated with $FeSO_4$ and diphenylamine indicator.

*Total nitrogen.* This was determined in the materials used to prepare the biomixture by the Nessler method (*Yuen & Pollard, 1954*). One gram of sample was dissolved in 20 mL of water, centrifuged at 2,000 rpm for 10 min, and filtered with Whatman paper # 42. To 20 mL of filtrate, three drops of polyvinyl alcohol and one mL of Nessler reagent were added. The absorbance was measured at 440 nm in a spectrophotometer (Thermo scientific Genesys 10S UV–VIS).

*pH.* To 10 g of sample, 50 mL of distilled water were added. The suspension was stirred (30 min, at 80 rpm) and left to stand for 10 min. The pH was measured with the potentiometer (Hanna HI98195) (*Pansu & Gautheyrou, 2007*).

*Moisture.* This was determined by gravimetry. 5 g of sample (previously dried at 110 °C for 2 h) was placed in a desiccator. Its weight was measured on an analytical balance until a constant weight was obtained (*Pansu & Gautheyrou, 2007*).

*Cation Exchange Capacity (CEC).* 5 g of soil was placed in a funnel with filter paper. 10 mL of 1N $CaCl_2$ was added to the sample and repeated five times. Then, 10 mL of ethanol was added five times, and the filtrate was removed. five mL of 1N NaCl was added five times, the filtrate (liquid) was stored and topped-off at 50 mL with 1N NaCl. Subsequently, 10 mL of the buffer solution pH 10 (67.5 mL of $NH_4Cl$ and 570 mL of $NH_4OH$ topped-off at 1,000 mL with water) was added. Five drops of KCN 2% solution and five drops of eriochrome black T indicator solution (0.1 g indicator and 1 g $NH_2OH$ HCl [hydroxylamine hydrochloride], diluted in 25 mL of methanol) were added. Finally, it was titrated with 0.05 N EDTA (versanate). The endpoint changed color from purple to blue (*Garman & Hesse, 1975*; *Pansu & Gautheyrou, 2007*). The CEC was calculated with Eq. (2):

$$\text{CEC (cmol(+) kg}^{-1}) = \frac{\text{volume EDTA (mL)} \times \text{EDTA N (eq g L} - 1) \times \text{CF}}{\text{sample weight (g)}} \tag{2}$$

where CF is the correction factor = (10 mL × 0.02 N)/mL EDTA (EDTA spend (mL) in the titration of 10 mL $CaCl_2$ 0.02 N solution). The CEC was expressed in the International System of units as cmol (+) kg$^{-1}$.

*Organic matter content.* This was determined by gravimetry using the loss-on-ignition method. 5 g of the sample was heated to 400 °C in an oven for 1 h. It was then placed in a desiccator and weighed on an analytical balance until a constant weight was obtained.

*Concentration of ETU.* To 2 g of soil or biomixture (previously dried), five mL of methanol-water solution (1:1) was added, vortex stirred for 2 min at 100 rpm, warmed at 70 °C (in a water bath) for 8 min and, treated in an ultrasonic bath for 15 min (Branson 2800). The sample was filtered under vacuum using a Büchner funnel and Whatman # 41 filter paper and centrifuged at 3,000 rpm for 15 min. The supernatant was filtered under vacuum using a 2 μm filter (Millipore). The sample was measured at 232 nm with a spectrophotometer (Thermo scientific Genesys 10S UV–VIS) (*Domínguez et al., 2021*).

*Microbial cultures.* For the isolation of bacteria and fungi, the plate dilution technique was used. The procedure was performed under aseptic conditions; 5 g of soil was weighed and placed in 45 mL of sterile water, and vortex stirred for 30 s. An aliquot of one mL was taken to prepare dilutions from $10^{-1}$ to $10^{-5}$. From each dilution, 100 μL were taken to inoculate Petri dishes with culture medium, in which the number of colonies was then evaluated. Bacteria were cultured for 24 h at 30 °C on 23 g $L^{-1}$ nutrient agar (MCD Lab) with 500 mL $L^{-1}$ soil extract, and pH adjusted to 5.6. The fungi were cultured for five days at 30 °C on 30 g $L^{-1}$ Sabouraud agar culture medium (MCD Lab) with a mixture of 500 mL $L^{-1}$ soil extract and 500 mg $L^{-1}$ chloramphenicol. The soil extract was prepared with a 1 kg $L^{-1}$ solution of Fluvisol soil, which was sterilized in an autoclave at 15 PSI, 121 °C for 15 min, filtered under vacuum (Whatman #42 filter paper), and the supernatant was topped-off at 1,000 mL with distilled water (*Atlas, 2005*; *Mueller, Bills & Foster, 2011*).

*Colony count.* The colony-forming units (CFU) were calculated through an analog colony counter with a magnifying glass. The CFU's of each treatment were calculated with Eq. (3):

$$\text{CFU mL}^{-1} = \frac{\text{Colonies counted} \times \text{reciprocal of the dilution}}{\text{Added volume (0.1 mL)}} \quad (3)$$

*Microbial cell count.* The procedure was repeated for each of the dilutions of soil samples previously prepared. A 10 μL drop of the corresponding dilution was placed in the Neubauer chamber. The number of bacterial cells and the number of fungal cells were observed by the optical microscope at magnifications of 100X and 40X, respectively. The number of microbial cells was calculated with Eq. (4):

$$\text{cells mL}^{-1} = \frac{\text{cell count} \times \text{reciprocal of the dilution}}{\text{area} \left(0.2 \text{ mm}^2\right) \times \text{chamber depth (0.1 mm)}} \quad (4)$$

*Experimental design and analysis of results.* The effect of EBDC on biomixture was analyzed. Unpolluted biomixture, polluted soil, and unpolluted soil were used as controls with three replicates per treatment. The experimental design was completely randomized. Response variables were evaluated and plotted over a 3-month kinetic with measurements every 14 days. Differences between treatments were analyzed using ANOVA and the Tukey-Kramer HSD test (honestly significant difference) with a significance level $\alpha = 0.05$. Spearman's multivariate statistical analysis was performed to describe the relationships between the variables studied. The statistical analysis was performed using the JMP 11.0.0 statistical software (Statistical Analysis System SAS®, 2014).

## RESULTS

### Kinetics of physicochemical parameters

The physicochemical variables were evaluated in the experimental units with soil and biomixture for 84 days. The variables considered were pH, moisture, CEC, organic matter content, and ETU concentration.

The pH tended to increase from slightly acidic to alkaline in all treatments. The pH varied from 5.9 to 7.9 in the polluted biomixture throughout the experiment. Slightly lower than in the unpolluted biomixture, from 6.3 to 8.5. However, the soil pH was lower than that observed in the biomixtures. It was from 5.0 to 6.4 in polluted soil and unpolluted soil from 5.3 to 6.9. There was no significant difference ($p < 0.05$) between the levels (surface, middle, and bottom) of each of the treatments (Fig. S1).

The moisture was significantly higher at the bottom compared to the middle and surface levels in both biomixtures and soils. In general, biomixtures had higher moisture than soils, with no difference between pesticide treatments. The lowest values were found in the surface level (30%, soil day 35) and the highest in the bottom level (70%, biomixture day 28). Moisture was on average 49% in polluted biomixtures and 52% in unpolluted biomixtures, while it was 44 and 47% with polluted and unpolluted soils (Fig. S2).

The CEC was higher in biobeds with soil as substrate; it was initially close to 62 cmol(+) kg$^{-1}$ and at the end of the experiment 86 cmol(+) kg$^{-1}$ on average. CEC was lower in biomixtures samples, close to 40 cmol(+) kg$^{-1}$ at the begging and 67 cmol(+) kg$^{-1}$ at the end of the experiment. There was no significant difference in column levels throughout the experiment in each treatment (Fig. S3).

Organic matter in polluted and unpolluted biomixtures were on average 20% and 18%, respectively. While in the polluted and unpolluted soils, it was 5% and 6%, respectively. The organic matter content was 3 to 4 times higher in the biomixtures than in the soils throughout the experiment. Organic matter decreased in the polluted biomixtures gradually from 23 to 16%. This decrease was slightly higher in the unpolluted biomixture. There was no significant difference in organic matter content between the levels (surface, middle, and bottom) of each treatment (Fig. S4).

At the beginning of the experiment, the highest ETU concentration occurred at the surface level of biomixtures and soils with 69 and 84 mg kg$^{-1}$ d. w., respectively. The concentration of the pesticide in the biomixtures decreased continuously at the surface level. There was an increase at 14 days in the intermediate level, while in the bottom at 14 and 28 days. On average, the pesticide concentration in biomixtures was 19 mg kg$^{-1}$ and in soils 30 mg kg$^{-1}$. After 84 days, it was found that in the biobeds with polluted biomixture, the concentration of ETU decreased to 5.0 mg kg$^{-1}$ in the surface level, 5.3 mg kg$^{-1}$ in the middle, and 1.6 mg kg$^{-1}$ in the bottom (Fig. 1). At the end of the experiment, the decrease of ETU in the polluted biomixtures was significant in all three levels of the column, while in the polluted soil, the decrease was smaller (Table 2).

### Kinetics of biological parameters

The biological variables were evaluated in the simulated biobeds with soil and biomixture for 84 days. The variables considered were microbial colonies and direct cell counts.

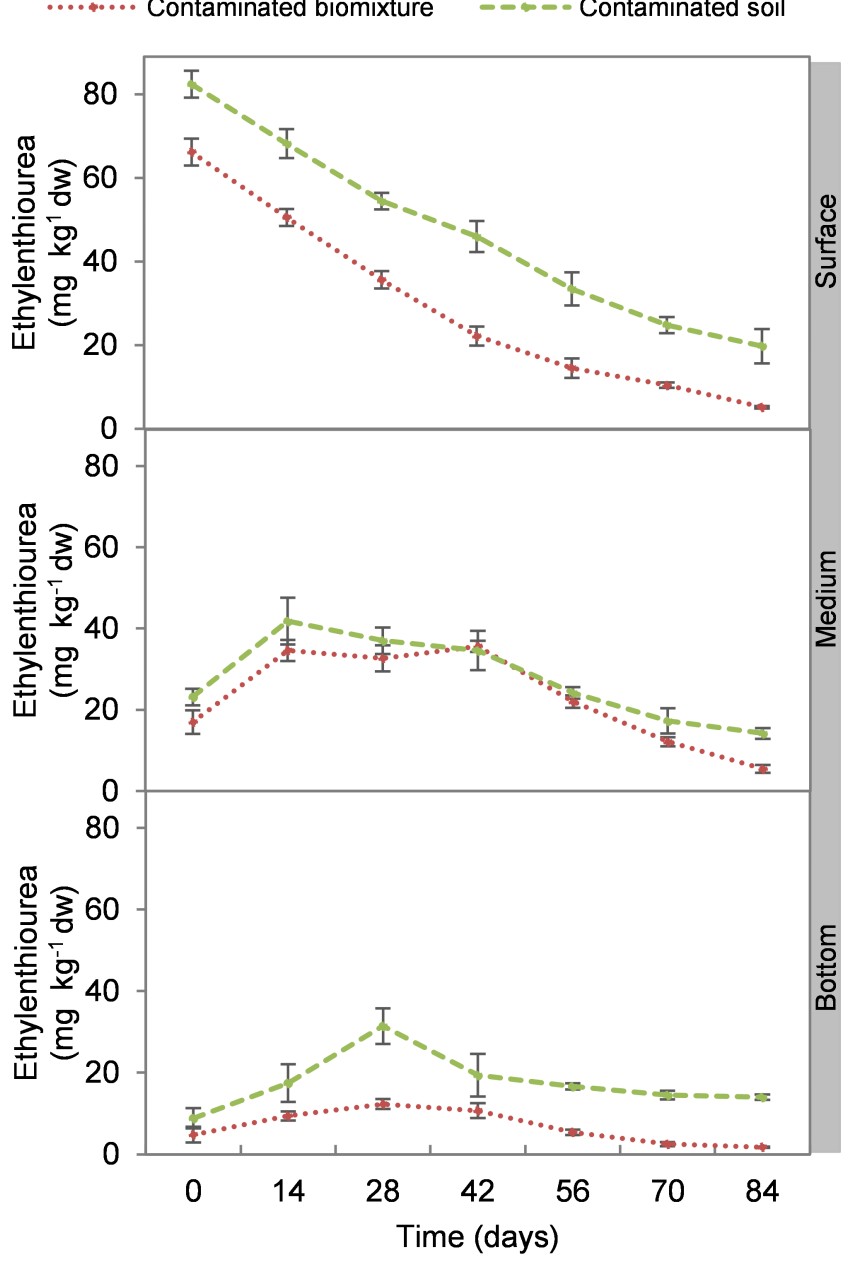

**Figure 1** **Kinetics of ethylenethiourea degradation at three depths in a biobed with soil and biomixture.** Error bars represent the standard deviation of three replications.

The addition of the pesticide in the biobeds significantly reduced microbial colonies in the treatments. The most significant impact was observed at the surface level of the biomixtures. It was found that the pesticide reduced bacterial colonies from $10 \times 10^7$ to $24 \times 10^6$ CFU mL$^{-1}$ and fungi from $36 \times 10^4$ to $40 \times 10^3$ CFU mL$^{-1}$ in the first seven days of exposure to the pesticide. In the middle and bottom levels of the biomixtures, a decrease in colony counts was observed on day 14. However, the number of bacterial
**Table 2  Comparison of initial and final values after 85 days of the microbial colony and cell counts in soil and biomixture polluted with ethylenethiourea.**

| Treatment | Level | Ethylenethiourea (mg L⁻¹) | Bacterial colonies (CFU mL⁻¹ × 10⁶) | Fungal colonies (CFU mL⁻¹ × 10⁴) | Bacterial count (cell mL⁻¹ × 10⁴) | Fungal count (cell mL⁻¹ × 10²) |
|---|---|---|---|---|---|---|
| Experiment start (Day 1) | | | | | | |
| Biomixture | Surface | 0.0 ± 0.0 h | 100 ± 4 a | 25 ± 4 bcd | 285 ± 18 ab | 75 ± 5 ab |
|  | Medium | 0.0 ± 0.0 h | 105 ± 2 a | 34 ± 2 a | 315 ± 30 a | 58 ± 7 bcd |
|  | Bottom | 0.0 ± 0.0 h | 108 ± 4 a | 34 ± 2 a | 265 ± 20 ab | 80 ± 10 a |
| Contaminated biomixture | Surface | 66.2 ± 3.2 b | 28 ± 4 fgh | 6 ± 2 ghi | 88 ± 12 cdef | 8 ± 2 jk |
|  | Medium | 17.0 ± 2.8 de | 100 ± 8 a | 34 ± 2 a | 250 ± 22 b | 40 ± 5 defg |
|  | Bottom | 4.8 ± 1.9 fgh | 105 ± 8 a | 34 ± 2 a | 235 ± 35 b | 33 ± 10 efghi |
| Soil | Surface | 0.0 ± 0.0 h | 46 ± 2 de | 14 ± 2 efg | 81 ± 7 cdefg | 18 ± 2 hijk |
|  | Medium | 0.0 ± 0.0 h | 53 ± 2 cde | 17 ± 2 def | 103 ± 7 cdef | 21 ± 5 fghi |
|  | Bottom | 0.0 ± 0.0 h | 56 ± 8 cde | 21 ± 2 bcde | 85 ± 10 cdefg | 21 ± 2 fghijk |
| Contaminated soil | Surface | 82.4 ± 3.2 a | 6 ± 2 i | 2 ± 2 i | 13 ± 2 i | 3 ± 2 k |
|  | Medium | 23.1 ± 2.0 c | 28 ± 4 fgh | 10 ± 2 fghi | 68 ± 7 defgh | 11 ± 2 jk |
|  | Bottom | 8.8 ± 2.4 f | 56 ± 8 cde | 21 ± 2 bcde | 96 ± 15 cdef | 18 ± 2 hijk |
| Experiment end (Day 84) | | | | | | |
| Biomixture | Surface | 0.0 ± 0.0 h | 62 ± 6 bcd | 24 ± 0 bcd | 108 ± 17 cde | 63 ± 2 abc |
|  | Medium | 0.0 ± 0.0 h | 74 ± 2 b | 29 ± 1 ab | 128 ± 15 c | 71 ± 16 ab |
|  | Bottom | 0.0 ± 0.0 h | 74 ± 8 b | 26 ± 2 abc | 115 ± 20 cd | 63 ± 10 abc |
| Contaminated biomixture | Surface | 5.1 ± 0.3 fg | 62 ± 8 bcd | 17 ± 2 def | 105 ± 18 cdef | 35 ± 10 efgh |
|  | Medium | 5.4 ± 0.9 fg | 65 ± 6 bc | 21 ± 4 bcde | 105 ± 22 cdef | 45 ± 5 cde |
|  | Bottom | 1.8 ± 0.1 gh | 65 ± 8 bc | 20 ± 4 cde | 105 ± 18 cdef | 41 ± 5 def |
| Soil | Surface | 0.0 ± 0.0 h | 41 ± 4 efg | 17 ± 2 def | 51 ± 16 fghi | 13 ± 2 ijk |
|  | Medium | 0.0 ± 0.0 h | 45 ± 4 def | 24 ± 4 bcd | 60 ± 13 efghi | 23 ± 5 fghijk |
|  | Bottom | 0.0 ± 0.0 h | 49 ± 6 cde | 26 ± 2 abc | 58 ± 12 efghi | 23 ± 7 fghijk |
| Contaminated soil | Surface | 19.7 ± 4.1 cd | 16 ± 4 hi | 5 ± 2 hi | 26 ± 7 hi | 20 ± 5 ghijk |
|  | Medium | 14.2 ± 1.3 e | 24 ± 4 ghi | 9 ± 2 fghi | 31 ± 5 ghi | 21 ± 5 fghijk |
|  | Bottom | 14.0 ± 0.6 e | 28 ± 4 fgh | 13 ± 2 efgh | 31 ± 10 ghi | 28 ± 2 efghij |

**Notes.**
*The letters in the columns indicate statistically significant differences between treatment ($\alpha = 0.05$), values not connected by the same letter are different.

and fungal colonies stabilized from day 42 days in all three levels. There was a significant recovery after 84 days compared to the unpolluted control. In the case of pesticide-polluted soil, the reduction in the number of bacterial colonies was higher, from $64 \times 10^6$ to $40 \times 10^5$ CFU mL⁻¹, and in fungi from $28 \times 10^4$ to $40 \times 10^3$ CFU mL⁻¹); this decrease was maintained throughout the kinetics study period at all three levels of the column (Figs. 2 and 3). After 84 days, there was no statistically significant difference between the number of bacterial cells in the polluted biomixture compared to the unpolluted biomixture in the three levels of the column. However, in fungal cells, the values are slightly lower in the polluted biomixture (Table 2).

Analogous to the colony count, the microbial cell count showed a significant impact due to the toxic effect of EBDC. The count of bacterial cells in the biomixture before the pollution was approximately $30 \times 10^5$ and decreased to $40 \times 10^4$ cells mL⁻¹, while in fungi, it decreased from $95 \times 10^2$ to $50 \times 10^1$ cells mL⁻¹. In pesticide-polluted soil, bacterial cells

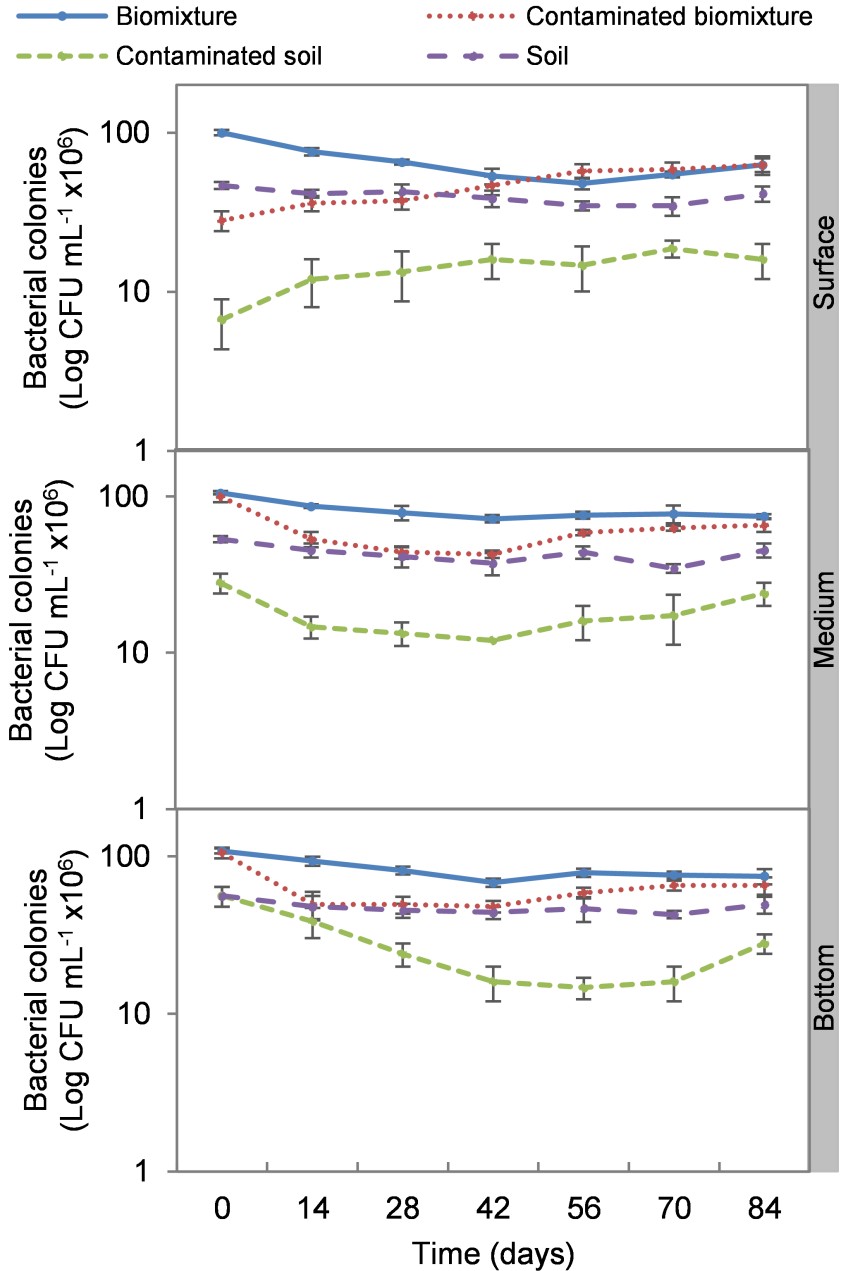

**Figure 2** **Kinetics of bacterial colonies at three depths in biobed polluted with Mancozeb (1,000 mg L$^{-1}$).** Error bars represent the standard deviation of three replications.

decreased to $10 \times 10^4$ cells mL$^{-1}$ and fungi even to none. The most significant reduction of microbial cells in polluted biomixture was observed at surface level on day 14. In the middle level, cells decreased from day 28 to 56, while in the bottom level, from day 56 to 60. After the decrease in the number of cells, the bacteria and fungi started showing marked recovery after about 42 days at the surface and middle level. However, fungi had a lower count in polluted samples throughout the experiment (Figs. 4 and 5).
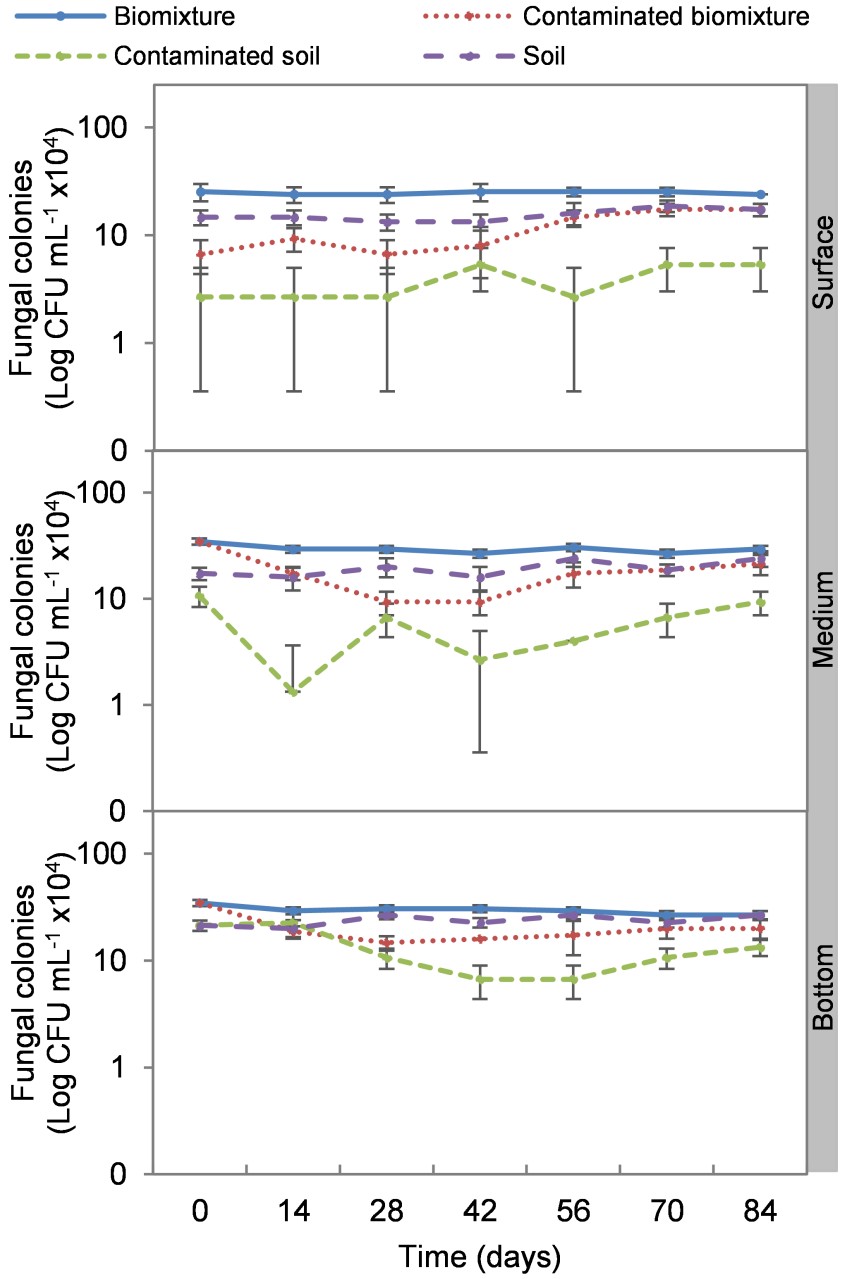

**Figure 3** **Kinetics of fungal colonies at three depths in biobed polluted with Mancozeb (1,000 mg L$^{-1}$).** Error bars represent the standard deviation of three replications.

Spearman's correlation analysis of variables was performed to identify the interaction of the parameters analyzed with microbial abundance (Table 3). It was found that there was a negative correlation between the number of microorganisms with the ETU concentration. The number of bacteria was significantly correlated with pH and CEC and ETU concentration, in that order. In contrast, the number of fungi had the highest significant correlation with ETU concentration, followed by pH and moisture. The highest

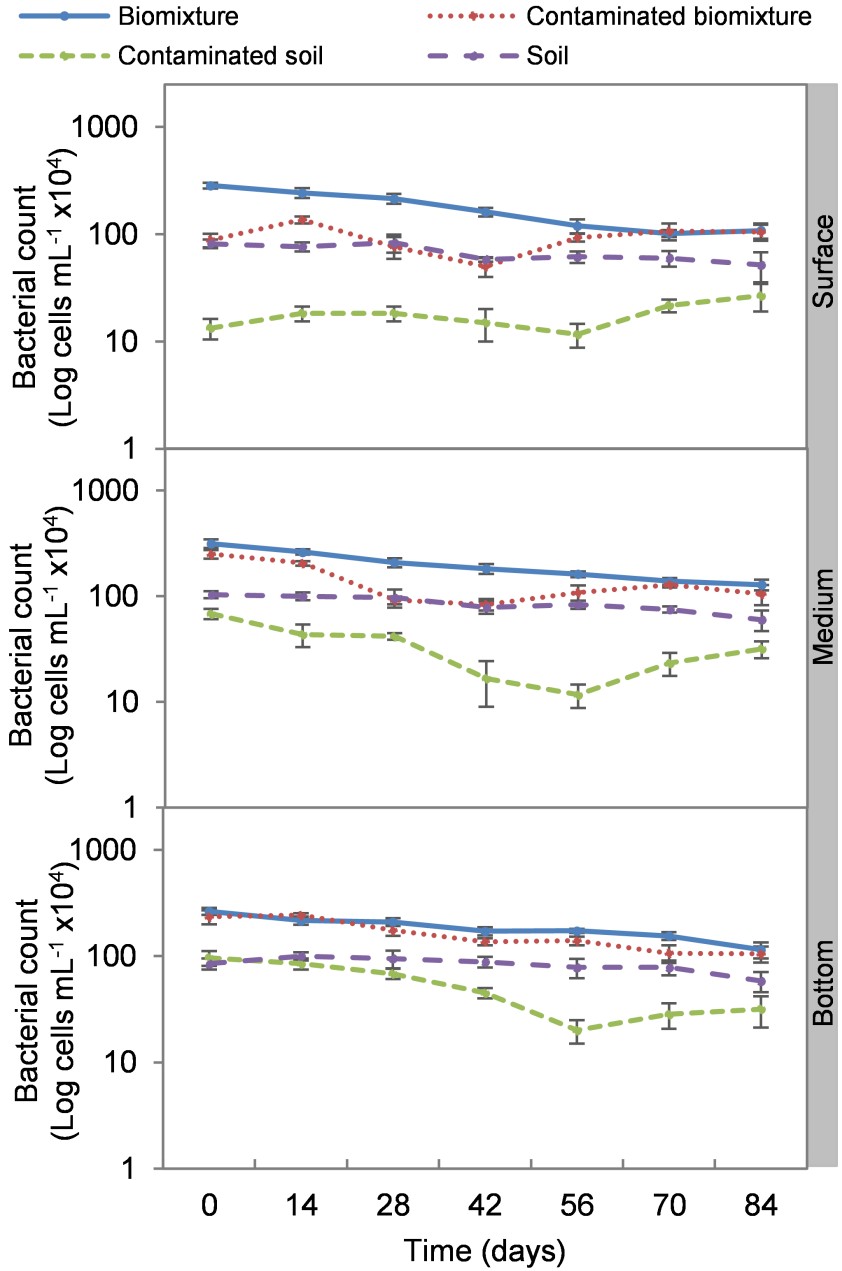

**Figure 4** **Kinetics of bacterial cells at three depths in biobed polluted with Mancozeb (1,000 mg L$^{-1}$).** Error bars represent the standard deviation of three replications.

correlation was ETU concentration with fungal cell count (0.77). The physicochemical parameters with the highest correlation were the organic matter with CEC (0.91).

## DISCUSSION

All treatments increased their pH because aerobic and anaerobic degradation of organic matter initially favored acidic processes, but as the biomixture matures, the pH tends

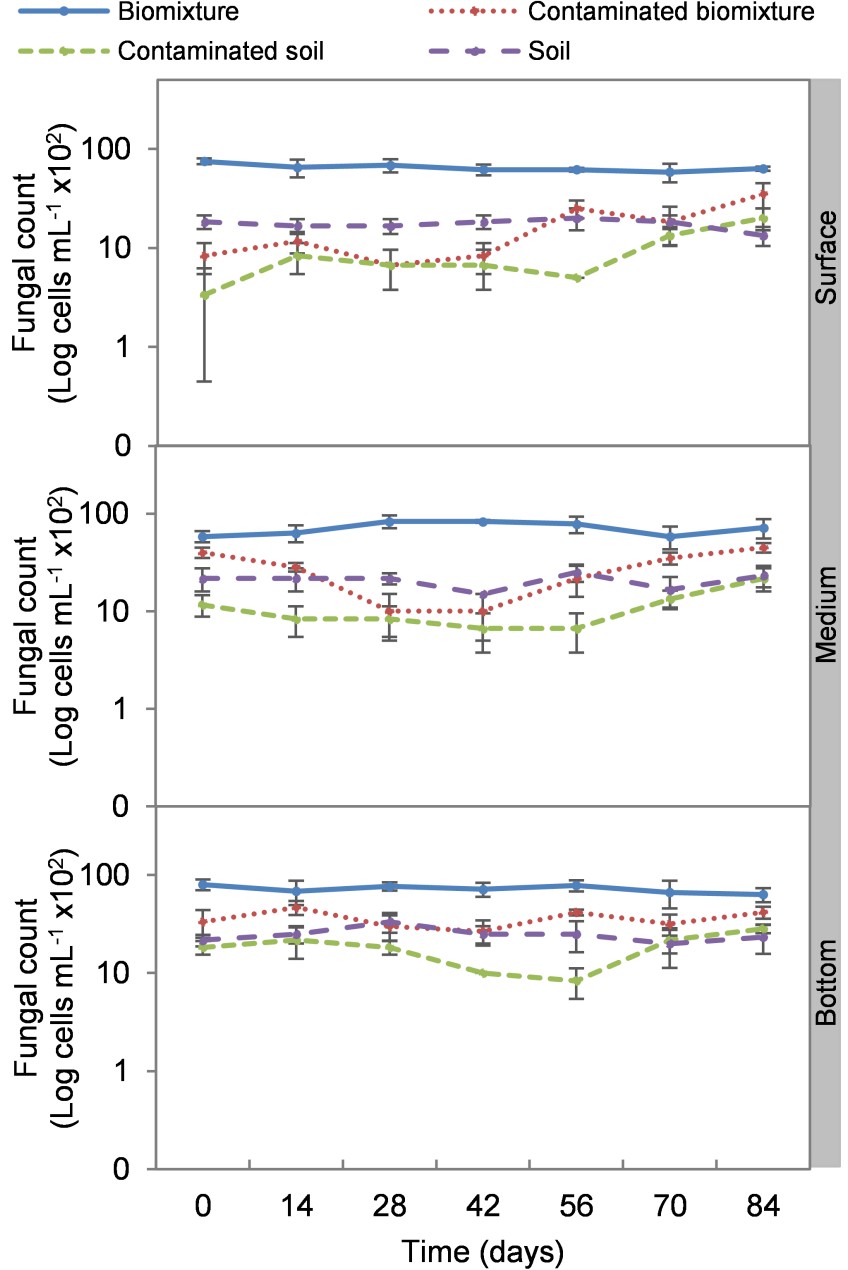

**Figure 5  Kinetics of fungal cells at three depths in biobed polluted with Mancozeb (1,000 mg L$^{-1}$).** Error bars represent the standard deviation of three replications.

to become alkaline (*Tortella et al., 2012*). The absence of differences in pH of the three levels of all treatments can be explained by the fact that the length of the column was not long enough for a gradient to form. The organic matter degradation due to microbial activity could explain the pH variations in treatment (*Tortella et al., 2012*). The most significant degradation of ETU occurred at near-neutral pH. According to *Castro et al. (2017)*, the pesticide oxamyl is rapidly hydrolyzed in soils with neutral pH, whereas it is

**Table 3 Correlation matrix between physicochemical and biological parameters in biomixtures and soils polluted with ethylenethiourea.** A ρ value with a negative sign indicates a negative correlation. Prob > |ρ| indicates the probability that the correlation is significant.

| Variable x | Variable y | ρ de Spearman | Prob > ρ |
|---|---|---|---|
| pH | Bacterial colonies | 0.77 | <.0001 |
| pH | Fungal colonies | 0.59 | <.0001 |
| pH | Bacterial cells | 0.59 | <.0001 |
| pH | Fungal cells | 0.61 | <.0001 |
| CEC | Bacterial colonies | −0.62 | <.0001 |
| CEC | Fungal colonies | −0.40 | <.0001 |
| CEC | Bacterial cells | −0.71 | <.0001 |
| CEC | Fungal cells | −0.24 | <.0001 |
| OM | Bacterial colonies | 0.66 | <.0001 |
| OM | Fungal colonies | 0.49 | <.0001 |
| OM | Bacterial cells | 0.76 | <.0001 |
| OM | Fungal cells | 0.28 | 0.001 |
| Moisture | Bacterial colonies | 0.62 | <.0001 |
| Moisture | Fungal colonies | 0.61 | <.0001 |
| Moisture | Bacterial cells | 0.55 | <.0001 |
| Moisture | Fungal cells | 0.46 | <.0001 |
| ETU | Bacterial colonies | −0.67 | <.0001 |
| ETU | Fungal colonies | −0.73 | <.0001 |
| ETU | Bacterial cells | −0.50 | <.0001 |
| ETU | Fungal cells | −0.77 | <.0001 |
| CEC | pH | −0.56 | <.0001 |
| ETU | pH | −0.58 | <.0001 |
| OM | pH | 0.57 | <.0001 |
| OM | CEC | −0.91 | <.0001 |
| CEC | Moisture | −0.26 | 0.003 |
| ETU | Moisture | −0.35 | <.0001 |
| OM | Moisture | 0.34 | <.0001 |
| Moisture | pH | 0.25 | 0.021 |
| ETU | CEC | 0.04 | 0.593 |
| ETU | OM | −0.10 | 0.244 |

**Notes.**
CEC, cation exchange capacity; ETU, ethylenethiourea; OM, organic matter.

slowly degraded in alkaline soils and with difficulty in acidic soils. According to *Vareli et al. (2018)*, the pH influences the sorption and mobility of pesticides. It was found that the biomixture alkalinization coincides with the ETU mobility from the surface to the bottom. Also, microbial populations are selected by the pH range of the biomixture. The increase in pH contributed to the decrease in the number of fungi. According to *Vischetti et al. (2007)* a slightly acidic pH may favor fungal activity, while the alkaline pH favors bacterial activity.

The cationic exchange capacity depends on the amount of clay and organic matter in the biomixture and soil (*Benito et al., 2005*). A soil with elevated CEC has a higher sorption capacity for pesticides (*Adak et al., 2020*; *Karanasios et al., 2010*). However, if an excess of

solutes saturates the sorption sites, the soil loses its sorption capacity (*Li et al., 2006*). It has been described that the CEC suitable for the degradation of pesticides should be lower than 60 cmol(+) kg$^{-1}$ (*Domínguez et al., 2021*). The CEC in the biomixtures allowed the ETU sorption in the middle and bottom of the biobeds until degradation.

The biomixtures retained adequate moisture throughout the experiment (35–65%); however, the values varied considerably at the different depth levels in the biobeds. The columns formed a moisture gradient because the high temperatures at the site favored the evaporation of the substrates. According to *Coppola et al. (2007)* the soil moisture content considered adequate for aeration and optimal microbial activity is 60%. However, it is crucial to avoid water saturation of the biomixture, as this would negatively affect the biodegradation process, with the risk of pesticides migrating out of the biobed (*Torstensson, 2000*). Like pH, the humidity of the middle and bottom levels may have been a factor influencing the mobility of ETU from the surface to the bottom of the biobeds since ETU is highly soluble in water (*Ruiz Suárez et al., 2013*).

*Diez et al. (2017)* considered that the appropriate organic matter content for pesticide degradation in the biomixture should be higher than 30%, which was not achieved in the experimental units of the experiment. This could be because the biomixture was not made from peat or compost, although the results were satisfactory in pesticide degradation. However, the organic matter content was higher than commonly found in soils of 1–6% (*Vischetti et al., 2008*). The treatments polluted with EBDC had a lower reduction in organic matter, possibly due to the toxic effect of the pesticide and subsequent reduction in microbial activity. There was no difference between the three levels of the column in all the treatments; this could be because the column length was not long enough to form a gradient. A high content of carbon-rich organic matter is essential because it increases the sorption capacity of the biomixture, preventing the formation of toxic leachates (*Kravvariti, Tsiropoulos & Karpouzas, 2010*). It has been described that biomixtures could have a pesticide retention capacity of up to 85% higher than most soils, so in biobeds, xenobiotics were retained in the upper layers and migrated slowly to the lower levels (*Delgado, Nogales & Romero, 2017*). The biomixture used was made up of 33% banana stem as an organic matter source. This material is rich in lignin (17%), cellulose (50%), and hemicellulose (15%) (*Abdullah, Sulaiman & Taib, 2013*). The biomixture should have a high lignin content so that the fungi produce the ligninolytic enzymes (laccases and peroxidases) that degrade the organic complexes (*Delgado, Nogales & Romero, 2017*; *Romero, Delgado & Nogales, 2019*). Lignocellulosic materials also supply carbon and nitrogen required for microbial growth (*Jia et al., 2017*; *Romero, Delgado & Nogales, 2019*). In this respect, the banana stem consists of up to 74% of its dry weight of organic carbon (*Abdullah, Sulaiman & Taib, 2013*), while the biomixture had 48% carbon. According to *Castro et al. (2017)*, the decrease of the carbon content in the biomixture causes a reduction of respiration and microbial activity; this may indicate the loss of the pesticide's degradation capacity or aging of the biomixture.

The initial concentration of the pesticide in this research was 1,000 mg L$^{-1}$, which can be considered high since most studies evaluating the degradation of pesticides in biobeds have used concentrations lower than 200 mg L$^{-1}$. In the literature review, only

three studies with similar concentrations were found. In the research conducted by *Gao et al. (2015)*, imidacloprid (1,000 mg L$^{-1}$) was degraded with a biomixture prepared with wheat straw, spend mushroom, and soil (2:1:1 v). *Perruchon et al. (2015)* reported 80% degradation of o-phenylphenol (1,000 mg L$^{-1}$) in 37 days in polluted soil. While, *Lescano et al. (2020)* found that a 90% degradation of glyphosate after 90 days (1,000 mg kg$^{-1}$) with a biomixture prepared with river residues, alfalfa, and wheat straws.

EBDC is poorly soluble (16 mg L$^{-1}$) and has a high soil adsorption coefficient ($K_{oc}$ = 363-2,334 cm$^3$ g$^{-1}$), however, ETU has a low soil sorption coefficient ($K_{oc}$ = 34–146 cm$^3$ g$^{-1}$) and is highly soluble (20,000 mg L$^{-1}$ 30 °C) (*Mackay, Shiu & Lee, 2006*). Therefore, ETU is a highly mobile metabolite upon contact with water. This mobility may explain the increase in ETU concentration in the middle and bottom levels of the biobeds from day 14 to 42; namely, the pesticide gradually migrated from the surface to the bottom in polluted biomixture due to higher humidity in the middle and bottom levels. The mobility of pesticides in biobeds will depend on the soil's sorption capacity, water solubility, and pH. Highly soluble pesticides with low sorption capacity tend to move through the soil, which decreases the residence time and the chances of being degraded by microorganisms (*Vareli et al., 2018*).

EBDC is known to degrade to ETU by photolysis in two days at 30 °C with normal atmospheric oxygen levels. On the other hand, ETU has a half-life of 1–9 days by photolysis (*Nikunen et al., 2000*). However, it has been reported that EBDC concentration >20 mg L$^{-1}$ in the soil can take 90–100 days to mineralize (*Cruickshank & Jarrow, 1973*). This report coincides with the degradation time found in the present research of 84 days. The ETU concentration at the end of the kinetics study period was close to the maximum residue limit established for food (tomato) of 2 mg kg$^{-1}$ day$^{-1}$ by the FAO (*Atuhaire et al., 2017*) and close to the median lethal dose defined for crustaceans (*Daphnia magna*) and fish (*Salmo gairdneri*) of 1.3 and 1.9 mg L$^{-1}$ respectively (*Nikunen et al., 2000*).

The pesticide addition in biomixtures caused a reduction of 72% in bacterial and 73% in fungal colonies at the surface level at the experiment's beginning. The pesticide toxicity was not observed in the middle and bottom levels until day 14 and then remained constant, possibly due to the mobilization of ETU from the surface to the bottom. The recovery of microbial populations on day 42 can be attributed to the reduction of toxicity at all three levels of the biomixture. At that time, the ETU concentration at the surface was 22 mg kg$^{-1}$, in the middle 35 mg kg$^{-1}$, and at the bottom 10 mg kg$^{-1}$. These values are lower than the half-maximal effective concentration ($EC_{50}$) calculated for microorganisms of 38 mg kg$^{-1}$ (*Van Leeuwen et al., 1985*).

Bacteria showed better adaptability to high pesticide concentrations than fungi. At the end of the experiment, the bacterial colony count in polluted and unpolluted biomixture was similar. The case of fungi had a significant reduction in the number of colonies between days 42 and 56. On the other hand, the reduction of colonies in polluted soil was very significant, 93% in bacteria and 85% in fungi. Results similar to those of this research were reported by *Diez et al. (2017)* on the degradation of 40 mg kg$^{-1}$ atrazine with a biomixture of wheat straw, peat, and soil. It was found that bacteria and fungi were strongly affected by atrazine, but microbial populations recovered after 40 days. It was

concluded that pesticide presence might have stimulated the growth of fungi capable of degrading the toxic compound. Also, it was considered that bacteria and actinobacteria could be associated with fungal populations in mineralizing the toxic compound in biobeds.

Few studies have evaluated the effect of pesticides on colony microbial counts. For example, *Tortella et al. (2013)* evaluated the effect of three doses of atrazine on the microbial population in a Swedish biomixture for 60 days. There was no difference in the number of bacterial and actinomycetes colonies compared to the control. The colony count was in the range of $14 \times 10^6$ to $45 \times 10^6$ CFU $g^{-1}$ in bacteria and $15 \times 10^5$ to $30 \times 10^5$ CFU $g^{-1}$ in actinomycetes. The number of fungal colonies decreased significantly compared to the control, with values from $21 \times 10^4$ to $75 \times 10^4$ CFU $g^{-1}$. It was concluded that fungi were more sensitive than bacteria and actinomycetes to atrazine. In comparison, *Góngora et al. (2020)* analyzed an inoculated biomixture with *Ochrobactrum* spp. and *Pseudomonas citronellolis* and polluted with 2,4-dichlorophenol, carbofuran, diazinon, and glyphosate (50 mg $L^{-1}$ each). Pesticides were degraded in 10 days, and the bacteria increased from $32 \times 10^6$ to $85 \times 10^6$ CFU $g^{-1}$ in five days.

The organic matter content of the biomixtures supported a higher microbial population compared to soils, which contributed to pesticide degradation. However, the microbial population may decrease to a level that may not contribute to pesticide degradation. In the kinetic study, the most significant decrease in microbial population coincided with the highest ETU concentration in the three levels of the polluted biomixture at 42 days with 42 to $48 \times 10^6$ CFU $mL^{-1}$ in bacteria and $80 \times 10^3$ to $16 \times 10^4$ CFU $mL^{-1}$ in fungi. This value in the polluted biomixture was similar to that in the unpolluted soil, $38$–$44 \times 10^6$ CFU $mL^{-1}$ in bacteria, $13$–$22 \times 10^4$ CFU $mL^{-1}$ in fungi. Based on the above, it is possible to infer that the required bacterial population to achieve EBDC degradation must be greater than $40 \times 10^6$ CFU $mL^{-1}$ andthe fungal population greater than $80 \times 10^3$ CFU $mL^{-1}$.

The cell count had a behavior analogous to the colony count. The addition of the pesticide caused a 69% reduction in bacterial cells and an 89% reduction in fungal cells at the surface level in the first 14 days in the polluted biomixtures. In the middle and bottom levels, the reduction on microbial cells was greater following the increase in ETU concentration. The stabilization of the number of bacterial cells was noticeable from day 42 of monitoring and in fungi less significantly. The number of bacteria fully recovered after 84 days while fungi were slightly lower, which is related to the dissipation of the pesticide in the three levels of the biobed. The difference in response between bacteria and fungi could be due to the chemical nature of the pesticide and that the greater diversity of bacteria favors the selection and growth of tolerant species. Other studies have also found greater sensitivity of fungi than bacteria to some pesticides. For example, *Campos et al. (2017)* analyzed in a Swedish biomixture the iprodione degradation (90.9 mmol $kg^{-1}$). It was concluded that the addition of the pesticide caused a decrease in fungal species abundance, but bacteria and actinobacteria adapted quickly. *Elgueta et al. (2017)* evaluated the degradation of atrazine, chlorpyrifos, and iprodione (35 mg $kg^{-1}$). It was found that all microbial groups were affected in soil, while in biomixtures only fungi.

Some studies have used cell counting to evaluate biomixtures, *Goux et al. (2003)* report that a sterile Swedish biomixture polluted with atrazine (10 mg $g^{-1}$) was inoculated with

575 cell g$^{-1}$ of microbial soil consortia. After 28 days, it had a concentration of $15 \times 10^3$ cell g$^{-1}$. *Sniegowski et al. (2012)* estimated that the minimum number of cells needed to remediate a linuron 60 mg L$^{-1}$ in a Swedish biomixture is $45 \times 10^1$ bacterial cells g$^{-1}$.

The highest decrease of microbial cells was at 42 days with a range of $50 \times 10^4$ to $13 \times 10^5$ cells mL$^{-1}$ in bacteria and $80 \times 10^1$ to $26 \times 10^2$ cells mL$^{-1}$ in fungi. This value in the polluted biomixture was similar to that in the unpolluted soil, 58 to $88 \times 10^4$ cells mL$^{-1}$ in bacteria, 15 to $25 \times 10^2$ cells mL$^{-1}$ in fungi. Analogously to the minimum colony number, it can be stated that the number of cells that could be necessary for EBDC degradation should be higher than $50 \times 10^4$ cells mL$^{-1}$ in bacteria and $80 \times 10^1$ cells mL$^{-1}$ in fungi. This value can help to indicate if the biomixture is aged; in this case, it should be replaced by a new biomixture. The suggested minimum number of microbial colonies and cells should be limited to biomixtures with similar characteristics in organic matter content exposed to a warm-humid environment.

The correlation analysis found that the number of bacterial colonies and cells is mainly affected by pH and CEC, followed by ETU concentration. In contrast, the number of fungal colonies and cells is mainly affected by ETU concentration followed by pH and moisture. It can be said that the degradation of ETU is simultaneously influenced by microbial activity, pH, moisture, and organic matter content. The abundance of fungi showed a higher correlation and sensitivity than bacteria to ETU concentration so that they could be used as indicators of EBDC degradation in biomixtures. The results demonstrate that colony count and cell count can be used to monitor the pesticide degradation process in biobeds, but without neglecting the analysis of other variables such as organic matter content, pH, CEC, and pesticide concentration.

Few investigations have analyzed the interaction between physicochemical and biological parameters in biobeds. In the research conducted by *Góngora et al. (2017)* in evaluating 11 types of biomixtures polluted with a mixture of five pesticides, he found that the concentration of residual pesticide has a relatively significant negative correlation with pH, lignin, C/N ratio, and water holding capacity. At the same time, with organic matter and nitrogen content, the correlation was less significant. This research concurred in finding that ETU concentration correlates strongly with pH and moisture, and to a lesser extent, with organic matter content. On the other hand, *Góngora et al. (2018)* evaluated biomixtures polluted with atrazine, carbofuran, diazinon, and glyphosate (12.50, 0.23, 0.34, and 0.36 mg cm$^{-3}$) exposed to a tropical climate. Twenty-three species of *Archeobacteria*, 598 species of bacteria, and 64 species of fungi were identified. Their research results pointed out that the archeobacteria diversity was correlated with pH and carbon/nitrogen ratio. In contrast, the bacteria diversity was correlated with lignin and organic matter content, while the fungal diversity with lignin content and water holding capacity.

Evaluation of microbiological parameters is necessary to understand the degradation kinetics of pesticides in biobeds (*Vischetti et al. 2008*). The pesticides alter the distribution and abundance of the microbial population. The significant impact occurs immediately upon contact with the pesticide, but if the dose is not excessively high, microbial populations may recover in a short time (*Diez et al., 2017*). However, there will be less impact on biomixtures than in soil without organic amendments. This buffering of the toxic effect

of pesticides is attributed to an increase in the sorption capacity of the matrix, as well as to the use of nutrients that come from the degradation of organic substrates by the microorganisms (*Delgado-Moreno et al. 2019*; *Elgueta et al. 2017*; *Wang et al. 2014*). Thus, selecting appropriate materials will influence the abundance of microorganisms with the capacity to degrade pesticides (*Vareli et al. 2018*).

Colony and cell counting have advantages over other techniques for assessing microbial activity. Enzyme activity assays and genetic profiling require expensive equipment or reagents. Genetic profiling helps establish diversity but does not quantify abundance. Moreover, just because a species is identified in a genetic profile does not indicate that it is metabolically active. On the other hand, biomass determination does not allow differentiation between microbial groups. Finally, colony and cell counts are relatively easy to measure, do not require expensive equipment, and are sensitive to pesticide variations.

## CONCLUSIONS

The biomixture based on banana stem, mulch, and Fluvisol soil (50:25:25% v/v) supported the microbial activity necessary to degrade the EBDC pesticide. This made it possible to take advantage of local materials and ensure the degradation of contaminants. It was found that a dose of 1,000 mg $L^{-1}$ reduced the number of microbial colonies by 72%. The number of bacterial cells decreased by 69% and fungi by 89% on the surface. ETU diffused to the bottom of the biofield, altering microbial distribution and abundance. The time required by the microorganisms to stabilize their populations after exposure to the EBDC compound was approximately 42 days. After 84 days, significant degradation of ETU was achieved at all three levels of the biomixture, 1.6 mg $kg^{-1}$ at the bottom and slightly higher at the middle and surface (5.0 and 5.3 mg $kg^{-1}$). At the end of the experiment, the bacteria showed significant recovery from the toxic effect of EBDC, but not the fungi. It was found that there is a strong correlation between ETU concentration and fungal counts; therefore, they could be used as indicators of the degradation process.

From the microbial growth kinetics, it was possible to establish the minimum populations necessary for pesticide degradation. In terms of microbial colonies, it was $40 \times 10^6$ CFU $mL^{-1}$ in bacteria and $80 \times 10^3$ CFU $mL^{-1}$ in fungi. While in microbial cells, it was set at $50 \times 10^4$ cells $mL^{-1}$ in bacteria and $80 \times 10^1$ cells $mL^{-1}$ in fungi. The microbial count can be used to know if the biomixture should be replaced when its capacity to maintain microbial activity is exhausted.

From the experimental results obtained in this research, it can be concluded that it is appropriate to include the quantification of microbial populations to assess the effectiveness of pesticide degradation and the lifetime of the biomixture. In this regard, the microbial colony and cell counting techniques used in this experimental work were convenient due to their low cost, ease of measurement, and sensitivity to pesticide variations. The microbial count made it possible to identify pesticide mobility within the biobed, the time required for degradation, and whether the microbial population is sufficient to support new doses of pesticides. Based on the found results, it is recommended to continue the research in the following aspects:

- The identification of indigenous microorganisms with the potential to degrade specific pesticides.
- Establishing the biomixture useful lifetime under continuous application of the pesticide conditions.
- The mechanism of tolerance to pesticides by the microorganisms and metabolic processes involved in the degradation.

### Funding
This work was supported by the Consejo Nacional de Ciencia y Tecnología (Cátedras CONACyT'' Program, Project No. 240). The funders had no role in study design, data collection and analysis, decision to publish, or preparation of the manuscript.

### Grant Disclosures
The following grant information was disclosed by the authors:
The Consejo Nacional de Ciencia y Tecnología (Cátedras CONACyT'' Program, Project No. 240).

### Competing Interests
The authors declare there are no competing interests.

### Author Contributions
- Verónica I. Domínguez-Rodríguez conceived and designed the experiments, performed the experiments, authored or reviewed drafts of the paper, and approved the final draft.
- Eduardo Baltierra-Trejo conceived and designed the experiments, performed the experiments, analyzed the data, prepared figures and/or tables, authored or reviewed drafts of the paper, and approved the final draft.
- Rodolfo Gómez-Cruz analyzed the data, prepared figures and/or tables, authored or reviewed drafts of the paper, and approved the final draft.
- Randy H. Adams conceived and designed the experiments, analyzed the data, authored or reviewed drafts of the paper, and approved the final draft.

### Data Availability
The raw measurements are available in the Supplemental File.

### Supplemental Information
Supplemental information for this article can be found online at http://dx.doi.org/10.7717/peerj.12200#supplemental-information.

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
