# Peer review of "Microbial growth in biobeds for treatment of residual pesticide in banana plantations"

_PeerJ, doi:10.7717/peerj.12200_

## Round 0.1 · original submission · Major Revisions

Dear Dr. Batierra-Trejo,

After careful reading of the reviewers' comments, I must recommend a profound revision of your manuscript, both in content and format, before being suitable for acceptance.

Both reviewers agree on the relevance of the study, but manifest serious concerns on the methodology used, and the conclusions derived from the study.

I recommend paying special attention to the major concerns raised by the two reviewers, and the conceptual and methodological issues posted by reviewer two.

I deeply suggest attending to all the issues at your best before resubmission.

·

Basic reporting

This an interesting study about the possible application of bioded made with local material to treat wastewater polluted with the fungicide ethylenebisdithiocarbamate. However, this work has several points that must be deeply clarified and improved, for example, the aim proposed is not discussed appropriately in relation to the obtained results, there are a lot of figures but the discussion is very poor as well as the association to other studies made by other authors. If one of the aims was to study the microorganisms distribution in the biobeds the conclusions should refer to that.
Your introduction needs more detail. I suggest that you improve the description at lines 63- 75 to provide more information and justification for your study (specifically, you should expand upon the knowledge gap being filled).
The raw data were shared and the structure of the article conforms to an acceptable format of standard sections.

More details about my revision of the different section are below:
Introduction
Lines 63 to 75: There are works about microbial population and enzymatic assays on biobeds and these articles are not mention neither discussed. The studies show different techniques to determinate for example Bacterial abundance (by qPCR) or bacterial and fungi population. Same of them are:
-Fate and effect of imidacloprid on vermicompost-amended soils under dissimilar conditions: Risk for soil functions, structure, and bacterial abundance. Jean Manuel Castillo Diaz, Fabrice Martin-Laurent, Jérèmie Beguet, Rogelio Nogales, Esperanza Romero. Science of the Total Environment 579 (2017) 1111–1119
-Aging of biomixtures: Effects on carbofuran removal and microbial community structure. Víctor Castro-Gutierrez, Mario Masís-Mora, María Cristina Diez, Gonzalo R. Tortella, Carlos E. Rodríguez-Rodríguez. Chemosphere 168 (2017) 418-425
-Indigenous biobed to limit point source pollution of imidacloprid in tropical countries. Totan Adak, Bibhab Mahapatra, Harekrushna Swain, Naveenkumar B. Patil, Guru P. Pandi G, G. Basana Gowda, M. Annamalai, Somnath S. Pokhare, Sankari Meena K, P.
C. Rath, Mayabini Jena. Journal of Environmental Management 272 (2020) 111084.

In general, the way that the results are presented is not attractive. The best form could be, for example using a unique title: Physicochemical parameters evolution. This style would help to have a discussion more integrated.
Lines 200-203: the description of results should be more appropriate to the aims of the work. For example Figure 2 shows that the pH follows the same tendency in both substrates (soil and biomixture) with a slight increase with the incubation time. Also, the behaviour is similar for the three profundities.
This comment is for all parameters described. ¿Why the analysis with the profundity is avoided in almost all parameters?.
In Figure 5 is not necessary (and not correspond) to show the values corresponding to soil and biomixture no polluted. If the values are very low the authors should use an axe Y secondary on the right side.
I recommend see other similar studies to improve the presentation of the results, for example (see the type of figures used):

-Indigenous biobed to limit point source pollution of imidacloprid in tropical countries
Totan Adak, Bibhab Mahapatra, Harekrushna Swain, Naveenkumar B. Patil, Guru P. Pandi G , G. Basana Gowda , M. Annamalai , Somnath S. Pokhare , Sankari Meena K , P.
C. Rath, Mayabini Jena. Journal of Environmental Management 272 (2020) 111084.

Figure 7 and 9: For this type of figure is better to use the Y axe in the logarithm scale (see the last paper recommended).
Figure 8 is not necessary; the analysis can be only in the text with the corresponding r2. Possibly exist other correlation tests that can be applied, if other environmental factors can also simultaneously influence the abundance of microbial populations may be the more appropriate apply a test like PCA. See other studies, for example:
-Agricultural effluent treatment in biobed systems using novel substrates from southeastern Mexico: the relationship with physicochemical parameters of biomixtures. Virgilio René Góngora-Echeverría, Fabrice Martin-Laurent, Carlos Quintal-Franco, German Giácoman-Vallejos, Carmen Ponce-Caballero. Environ Sci Pollut Res, DOI 10.1007/s11356-017-8643-z

Experimental design

The work is within Aims and Scope of the Journal.

Section methods. The authors need to clarify the following:
Were the materials required for biomixture treated?, i.e, these were cut into pieces of determinate size or were used directly.
Line 100: why a concentration of 1000 mg/L was used?. Justify.
Lines 104-105: to add the different depths of biobed-column (surface, medium, and bottom) in cm.
Discussion
Line 246-248: It would be interesting found a relationship between physicochemical parameters, microbial population and compound degradation under a point of view statistic. There is a lot of information but must be rigorously analysed.
Lines 258-264: the paragraph describes other works but no explore the relationship with the results obtained in this work. Rewrite.
Line 266-271 and 273-279: Again, rewrite taking into account the owner results.
Line 281: Organic matter: something to say about the results of organic matter vs. profundities?
Lines 311 and 312: The sentence is not clear. Improve the idea: At 85 days of pesticide degradation, the deepest level’s concentration decreased to an extent acceptable for soil organisms growth, close to 1 mg L-1
The authors can give some explanation about why the concentration of ETU increases between 14 and 42 days in the medium and bottom of the biobed.
Lines 320 and 321: How is this conclusion obtained? The information can be seen in the figures?. Please, explain better the idea of the following sentence:
It was found that the number of cells that could be necessary for EBDC degradation should be greater than 100 x 105 cells mL-1 in bacteria and close to 20 x 103 cells mL-1 in fungi.

Validity of the findings

-The conclusions should mention something about the effect of profundity.
-The concentration (3.7 mg L-1) of the pesticide at 85 days: is an average value between the different profundities?.
Line 382 to 384: I do not think that colony and cell counting techniques are simple. I think that indirect technique like analyses of the enzymes (for example FDA) can be more simple.

Additional comments

Other comments:
Line 134: replace NH2OH by NH4OH
Lines 158 and 164: should say “soil or biomixture” since the assay was done for both materials.
Lines 197 and 198: this sentence and the Table 1 must be in Material and methods section after the description of the material.
For CEC the units used generally is meq/g. Why the reported values have a subindex: cmolc kg-1?
In Figure 5 the ET concentration increases at the medium and the bottom. Do the authors have any explanation?

·

Basic reporting

This manuscript describes a laboratory-scale biobed system for the treatment of pesticide EBDC from a banana plantation. While the aims and subject (crop and pesticide) of the study are of interest, the manuscript suffers from issues with readability, and there are important concerns related to the methodology employed and subsequent conclusions drawn.

The manuscript would benefit from review and contribution from a proficient English writer as there are several issues with grammar/syntax, flow and readability. For example, line 19-20: “frequently unused pesticide mixture (solution) is not disposed of properly” should be “unused pesticide mixture (solution) is frequently improperly disposed of…” and lines 46-67 include a dependent “because” statement that is not resolved. These are two of many examples.

The introduction is sparse with current literature describing biobed microbial community dynamics and treatment efficacy. Furthermore, lines 72-75 contain references and information that are not relevant to the study at hand.

The use of subheadings in the discussion (as used in Results) should be avoided and discussion of these topics integrated into a broader and flowing narrative consideration of the results.

While several physicochemical explanatory parameters of the biobed were investigated, the presentation of all this data is not entirely necessary and would be more suitable as supplementary material in an improved manuscript that included expanded analysis (see further comments below). In the figure legends of line graphs (Figs 1-7), replicate numbers and error bars are not described/explained.

Experimental design

While the research topic is broadly relevant as a novel biobed treatment application (i.e., banana crop and EDBC), the research questions is vaguely stated and is not appropriately interrogated with the methodology used. Understanding physicochemical parameters of a functioning biobed are certainly relevant for understanding overall system dynamics and with respect to monitoring, but this research stops short of characterizing the health, diversity, nature/characteristics and stability of the biobed microbial community and linkage between these communities and the treatment efficacy of EDBC in any way. Cell counts (direct or CFU) of microbial cells are not representative of the microbial population present or micorbial activity, merely those ones capable of growth on that particular medium. Indeed, it is demonstrated within the paper itself (Figure 8), that there is little to no correlation between any cell counts and pesticide concentration.

There are also concerns with the detail provided for the methodology section related to the setup of the biobed systems – (i.e., replication, experimental design, how samples were collected, irrigation). Furthermore, the biobed systems as described appear to be considerably small in size (50 x 9.5 cm) and received one pulse of pesticide rinsate inflow/input and thus applicability to full-scale, on-site systems is of questionable utility.

Lastly, recent/current investigation of biobed treatment systems employ microbial community profiling techniques (i.e., sequencing) to specifically characterize “microbial dynamics” of biobed treatment systems, which any cursory literature search would reveal. As conducted and presented, this paper does not deliver contemporaneous results that advance understanding of the field.

Validity of the findings

Within the presentation and discussion of results, the rationale for the experimental design is not well explained or connected and thus the impact the data has is not clear. As previously mentioned, the sole examination of bacterial counts and the use of small bioreactor systems severely hampers the ability to draw any conclusions as to the nature of “microbial dynamics” in the treatment of EDBC using banana compost.

The conclusive statements related to a specific number of microbial counts being required for EDBC degradation is not supported given the lack of proper controls and demonstration that the microbes contribute substantially to the degradation (other factors such as sorption, dilution, half-life degradation are not considered or examined), and by the authors own statistical analysis. Replication is not made clear throughout (i.e., in figure captions) and there is limited impactful discussion of their experimental design (i.e., the four treatments analyzed). The authors also discuss microbial species within biobeds that have been identified in other research and draw unsupported and inappropriate connection to the present study, as no similar analysis was conducted.

While the authors are certainly correct to state that is necessary to develop cost-effective means of biobed assessment, particularly in remote areas/situations that rely on local borrow-sources, however the conclusions that the authors understand the “minimum population of cells required for treatment” is not supported by current experiment. Further experimentation using appropriate controls, replication, and demonstration of biobed efficacy over biobed management lifecycle and/or continuous pesticide treatment is required in order to support the claims of understanding “microbial dynamics”.

Additional comments

While the overall objective of the study is sound and of utility to the broader field of investigation, the methodology used to investigate this crop/pesticide treatment, and the overall design of the experiment do not support the ultimate conclusions made. The authors must consider i) the role that other means of pesticide degradation (sorption, half-life, etc.) might play in the observed concentration reduction ii) the scale of their experimental system and the applicability it may have to large-scale, on-site systems iii) adjusting their experimental design such that several of the “future research” questions may indeed be answered (i.e., replacing materials, increased irrigation, continuous or pulse-pesticide application, sequencing of communities, and use of control biobeds/matrices).

---

## Round 0.2 · Minor Revisions

The reviewer has raised some minor comments on the manuscript that need your attention. Some of them are typos, others are doubts or suggestions.

Please respond to these queries in order to proceed with the acceptance of the manuscript.

·

Basic reporting

pass

Experimental design

pass

Validity of the findings

pass

Additional comments

I consider that the manuscript was deeply improved, the discussion now is more clear and it is well connected with the aims and the obtained results. I think that the results are valuable for monitoring the biobeds in the banana production because suggest bacterial and fungal populations (or cell count) to optimal working of the biomixture or for its replace. I agree with the authors when they say that the techniques used in this work were convenient due to their low cost and simplicity (no require expensive equipment or reagents). This is key factor to apply this technology in producing areas.
I have minor comments that annotated in PDF files (57885-v1).

---

## Round 0.3 · accepted · Accept

I have no more comments on the recent version of the manuscript. The responses fulfill the requirements posted by reviewers.